# Production and Characterization of Self-Assembled Virus-like Particles Comprising Capsid Proteins from Genotypes 3 and 4 Hepatitis E Virus (HEV) and Rabbit HEV Expressed in *Escherichia coli*

**DOI:** 10.3390/v16091400

**Published:** 2024-08-31

**Authors:** Tominari Kobayashi, Masaharu Takahashi, Satoshi Ohta, Yu Hoshino, Kentaro Yamada, Suljid Jirintai, Putu Prathiwi Primadharsini, Shigeo Nagashima, Kazumoto Murata, Hiroaki Okamoto

**Affiliations:** 1Division of Virology, Department of Infection and Immunity, Jichi Medical University School of Medicine, Shimotsuke 329-0498, Tochigi, Japan; kobayashi-tm@jichi.ac.jp (T.K.); mtaka84@jichi.ac.jp (M.T.); thiwik8@jichi.ac.jp (P.P.P.); shigeon@jichi.ac.jp (S.N.); kmurata@jichi.ac.jp (K.M.); 2Division of Structural Biochemistry, Department of Biochemistry, Jichi Medical University School of Medicine, Shimotsuke 329-0498, Tochigi, Japan; satoshi.ohta@jichi.ac.jp

**Keywords:** hepatitis E virus, capsid protein, virus-like particle, genotype, *Escherichia coli*, assembly in vitro

## Abstract

The zoonotic transmission of hepatitis E virus (HEV) genotypes 3 (HEV-3) and 4 (HEV-4), and rabbit HEV (HEV-3ra) has been documented. Vaccination against HEV infection depends on the capsid (open reading frame 2, ORF2) protein, which is highly immunogenic and elicits effective virus-neutralizing antibodies. *Escherichia coli* (*E. coli*) is utilized as an effective system for producing HEV-like particles (VLPs). However, research on the production of ORF2 proteins from these HEV genotypes in E. coli to form VLPs has been modest. In this study, we constructed 21 recombinant plasmids expressing various N-terminally and C-terminally truncated HEV ORF2 proteins for HEV-3, HEV-3ra, and HEV-4 in *E. coli*. We successfully obtained nine HEV-3, two HEV-3ra, and ten HEV-4 ORF2 proteins, which were primarily localized in inclusion bodies. These proteins were solubilized in 4 M urea, filtered, and subjected to gel filtration. Results revealed that six HEV-3, one HEV-3ra, and two HEV-4 truncated proteins could assemble into VLPs. The purified VLPs displayed molecular weights ranging from 27.1 to 63.4 kDa and demonstrated high purity (74.7–95.3%), as assessed by bioanalyzer, with yields of 13.9–89.6 mg per 100 mL of TB medium. Immunoelectron microscopy confirmed the origin of these VLPs from HEV ORF2. Antigenicity testing indicated that these VLPs possess characteristic HEV antigenicity. Evaluation of immunogenicity in Balb/cAJcl mice revealed robust anti-HEV IgG responses, highlighting the potential of these VLPs as immunogens. These findings suggest that the generated HEV VLPs of different genotypes could serve as valuable tools for HEV research and vaccine development.

## 1. Introduction

Hepatitis E virus (HEV) is the leading cause of acute viral hepatitis worldwide, accounting for over 3.3 million symptomatic cases of hepatitis annually and resulting in approximately 44,000 hepatitis E-related deaths [1,2]. Globally, seroprevalence studies have indicated that one in eight individuals has been infected with HEV, as evidenced by the presence of anti-HEV IgG antibodies [3]. Typically, HEV infection results in asymptomatic or self-limited disease; however, in immunocompromised patients, it can become chronic or be associated with extrahepatic manifestations, including neurological and renal disorders [4,5]. The mortality rate in the general population ranges from 0.5 to 4.0%, while in pregnant women, it can reach up to 30% [6,7]. The primary cause of the increasing number of autochthonous human infections in industrialized countries is zoonotic transmission of HEV [8].

HEV is a quasi-enveloped virus with an icosahedral capsid that encloses its viral genome [9]. The HEV genome consists of a single-stranded, positive-sense RNA, ranging from 6.6 to 7.3 kilobases (kb) in length, flanked by 5′ and 3′ untranslated regions (UTRs) [10,11]. It features a 7-methylguanine cap at the 5′ end and a poly(A) tail at the 3′ end, resembling host mRNA [12]. The HEV genome contains three open reading frames (ORFs: *ORF1–3*). *ORF1* encodes a large non-structural polyprotein with multiple functional domains involved in viral replication, such as methyltransferase, helicase, and RNA-dependent RNA polymerase [13]. *ORF2* and *ORF3* proteins are translated from a 2.2 kb bicistronic subgenomic RNA [14]. *ORF2* encodes a structural capsid protein (660 amino acids [aa]) that interacts with putative host receptors and is a major target for neutralizing antibodies; a secreted form of ORF2 does not prevent virus entry but interferes with antibody-mediated neutralization [15,16]. *ORF3* encodes a small multifunctional phosphoprotein (112–114 aa) that functions as an ion channel (viroporin) and is essential for virion morphogenesis and egress [17,18].

HEV is classified within the family *Hepeviridae*, which comprises two subfamilies: *Orthohepevirinae* and *Parahepevirinae*. The *Orthohepevirinae* encompasses four genera: *Paslahepevirus*, *Rocahepevirus*, *Chirohepevirus*, and *Avihepevirus* [10]. The genus *Paslahepevirus* comprises two species, *P. alci* and *P. balayani*. The species *P. balayani* includes eight viral genotypes, HEV-1 to HEV-8 [19]. HEV-1 and HEV-2 strains are exclusively found in humans and are responsible for large outbreaks of hepatitis E in developing countries. In contrast, HEV-3 and HEV-4 strains have been isolated from domestic pigs and wild boars, causing sporadic hepatitis E cases in both developing and industrialized countries [20,21]; HEV-5 and HEV-6 strains have been detected solely in wild boar populations in Japan [22,23,24,25]; HEV-7 infects dromedary camels [26], while HEV-8 infects Bactrian camels [27,28]. 

HEV-3 is further divided into at least fourteen subtypes (3a–3m and 3ra), and HEV-4 is divided into at least nine subtypes (4a–4i) [19]. Notably, HEV subtypes 3a, 3b, 3e, 3k, 4c, 4g, and 4i are actively circulating in Japan, with subtypes 3b and 4c being the most prevalent for each genotype. These subtypes are associated with zoonotic food-borne transmission in Japan [24,29,30,31,32]. Notably, HEV-3ra (rabbit HEV) strains have been isolated from both farmed and wild-caught rabbits, forming a distinct clade closely related to HEV-3 [33,34]. Recent studies have demonstrated that HEV-3ra is capable of infecting humans and causing hepatitis [35,36].

Recently, apart from HEV-3, HEV-3ra, HEV-4, and HEV-7, zoonotic infections of humans with rat HEV (*Rocahepevirus ratti* species, HEV-C1) have been documented. These infections have caused chronic infections in immunosuppressed individuals and acute hepatitis even in children [37,38,39,40]. Rat HEV has been isolated from wild rats globally [41,42,43], suggesting that rat HEV might be an underestimated source of human infection. The protein encoded by *ORF2* of rat HEV is 644 aa long, and 16 aa shorter than that of human HEV.

Vaccination-based prevention of HEV infection hinges on the highly immunogenic capsid protein, which elicits potent virus-neutralizing antibodies [44]. To date, various systems have been utilized to express the HEV capsid protein, including *Escherichia coli* (*E. coli*), insect cells, mammalian cells, and plants [45,46]. Among these, *E. coli* and the baculovirus-insect cell system are considered the most effective for producing virus-like particles (VLPs) [47,48,49,50,51,52]. In our previous study, we successfully expressed rat HEV ORF2 proteins in *E. coli* and characterized the self-assembled VLPs [53]. Despite advances in VLP production for HEV vaccination, exemplified by the recombinant VLP-based vaccine Hecolin® (HEV-1), available in China and Pakistan [51,54], research on the production of ORF2 proteins from various HEV genotypes in *E. coli* forming VLPs remains limited [45,46,53]. Notably, the construction of VLPs from HEV-3, including HEV-3ra (rabbit HEV), in *E. coli* has not been reported. In the present study, we expressed HEV ORF2 proteins from HEV-3 (subtype 3b), rabbit HEV (HEV-3ra), and HEV-4 (subtype 4c) in *E. coli* and characterized the self-assembled VLPs towards studies on the antigenicity, immunogenicity, pathogenicity, and epidemiology of zoonotic HEV and the future development of VLP-based vaccines for HEV prevention.

## 2. Materials and Methods

### 2.1. Plasmid Construction and Truncated HEV ORF2 Protein Expression

To construct plasmids expressing truncated HEV-3 ORF2 proteins of the JE03-1760F strain (subtype 3b) [55], nine expression vectors were developed. These vectors contained truncated N- and C-termini sequences (aa 12–606, 14–606, 61–606, 89–606, 107–606, 112–606, 117–606, 122–606, and 368–606 of *ORF2*) (Figure 1A). The nucleotide sequences corresponding to aa 12–606, 14–606, 61–606, 89–606, 107–606, 112–606, 117–606, 122–606, and 368–606 of *ORF2* were amplified via polymerase chain reaction (PCR) using the full-length JE03-1760F cDNA clone (AB437316) [56] as a template. The primer set comprised a sense primer with an introduced Nde I site (hHEV_G3 Nde-ORF2 Fw, corresponding to aa 12, 14, 61, 89, 107, 112, 117, 122, or 368 of *ORF2*) and an antisense primer with a BamH I site (hHEV_G3 ORF2-Bam Rev, corresponding to aa 606 of *ORF2*) (Table 1). KOD Plus DNA polymerase (TOYOBO, Osaka, Japan) was used for PCR amplification.

In addition, two expression plasmids for HEV-3ra ORF2 proteins of the rbIM223L strain [34] were constructed with truncated N- and C-termini (aa 112–606 and 368–606 of ORF2) (Figure 1A). The nucleotide sequences corresponding to aa 112–606 and 368–606 of *ORF2* were amplified by PCR utilizing an *ORF2* cDNA clone derived from the culture medium of the rbIM223L strain (LC775585) as a template. Prior to the amplification, T at nt 6686 within the template had been converted to C through inverted RT-PCR utilizing the rbHEV_G3 T6686C Fw and rbHEV_G3 T6686C Rev primers (Table 1) to eliminate the internal Nhe I sequence (CATATG), without altering the amino acid sequence. The primer set included a sense primer with an introduced Nde I site (rbHEV_G3 Nde-ORF2 Fw, corresponding to aa 122 or 368 of *ORF2*) and an antisense primer with a BamH I site (rbHEV_G3 *ORF2*-Bam Rev, corresponding to aa 606 of *ORF2*) (Table 1). KOD Plus DNA polymerase was used for amplification.

Furthermore, ten expression plasmids for HEV-4 ORF2 proteins of the HE-JF5/15F strain (subtype 4c) [58] were constructed with truncated N- and C-termini (aa 112–606, 114–606, 117–606, 366–606, 368–606, 370–606, 112–609, 117–609, 112–612, and 117–612 of *ORF2*) (Figure 1A). The nucleotide sequences corresponding to aa 112–606, 114–606, 117–606, 366–606, 368–606, 370–606, 112–609, 117–609, 112–612, and 117–612 of *ORF2* were amplified by PCR using the full-length HE-JF5/15F cDNA clone (LC775584) as a template. The primer set consisted of a sense primer with an introduced Nde I site (hHEV_G4 Nde-*ORF2* Fw, corresponding to aa 112, 114, 117, 366, 368, or 370 of *ORF2*) and an antisense primer with a BamH I site (hHEV_G4 ORF2-Bam Rev, corresponding to aa 606, 609, or 612 of *ORF2*) (Table 1). KOD Plus DNA polymerase was employed for PCR amplification.

The resulting amplicons were excised and ligated into the Nde I and BamH I sites of the pET-3a expression vector (Merck, Darmstadt, Germany) (Figure 1B). A total of 21 plasmids with an insert of desired length (Figure 1A) were confirmed through nucleotide sequencing. Sequencing was performed using the Applied Biosystems 3130xl Genetic Analyzer (Thermo Fisher Scientific, Waltham, MA, USA) in conjunction with the BigDye Terminator v3.1 Cycle Sequencing Kit (Thermo Fisher Scientific). A sequence analysis was performed using the Genetyx software program (version 13; Genetyx Corp., Tokyo, Japan).

Protein expression was carried out in *E. coli* strain BL21 (DE3) (New England BioLabs, Ipswich, MA, USA) using 100 mL of TB medium containing 12 g/L tryptone, 24 g/L yeast extract, 8 mL/L glycerol, 9.4 g/L K_2_HPO_4_, 2.2 g/L KH_2_PO_4_, and 0.1 mg/L ampicillin. The bacterial cultures were grown at 37 °C until reaching an optical density (OD600) of 1.0. Protein expression was then induced by the addition of 0.2 mM isopropyl β-D-1-thiogalactopyranoside (IPTG) and incubation at 25 °C overnight. The cultures were centrifuged at 2150× *g* for 20 min at 4 °C, and the resulting pellets were either used immediately for the formation of VLPs or stored at −80 °C until further use.

### 2.2. Purification of Recombinant HEV ORF2 Proteins

Bacterial cell pellets obtained from a 100 mL culture in TB medium were resuspended in a sonication buffer at pH 7.5, containing 10 mM NaH_2_PO_4_-2H_2_O, 10 mM Na_2_HPO_4_-12H_2_O, 100 mM NaCl, 0.5 mM EDTA, 1 mM dithiothreitol, 1 µg/mL RNase A, and 1 µg/mL DNase I. The suspension was then subjected to sonication on ice using an ultrasonic homogenizer (NR-50M; MICROTEC Co., LTD, Chiba, Japan) at 65% amplitude for 12 cycles of 1 min pulses. Following sonication, the lysate was incubated with 2% (*v*/*v*) Triton X-100 at 37 °C for 30 min. To separate the lysate into a supernatant fraction and a pellet fraction containing inclusion bodies, the mixture was centrifuged at 2150× *g* at 4 °C for 15 min. The resulting pellet was washed twice with the sonication buffer. The inclusion bodies were subsequently solubilized in 10–30 mL of 4 M urea buffer at pH 7.5, containing 10 mM NaH_2_PO_4_-2H_2_O, 10 mM Na_2_HPO_4_-12H_2_O, 100 mM NaCl, and 4 M urea, followed by sonication at 70% amplitude for 5–20 cycles of 40 s pulses. The solubilized proteins were clarified by centrifugation at 20,400× *g* for 5 min at 20 °C, and the resulting supernatant was collected. This supernatant was subsequently passed through 0.45 µm pore-sized microfilters (Sterivex-HV; EMD Millipore Corp., Billerica, MA, USA) to ensure further clarification. Eleven milliliters of the filtered supernatant were subsequently then subjected to gel filtration chromatography using fast protein liquid chromatography (FPLC) on a HiPrep 16/60 Sephacryl S-400 HR column (GE Healthcare Japan, Tokyo, Japan) at 4 °C, with a flow rate ranging from 1.0 to 1.4 mL/min. This process aimed to facilitate protein renaturation, assembly, and purification, with phosphate-buffered saline (PBS) employed as the running buffer. Fractions containing VLPs, each with a volume of 4 mL, were pooled, typically yielding three to four fractions.

### 2.3. SDS-PAGE and Western Blot Analysis

Various protein samples obtained in this study—including the supernatant fraction, the pellet fraction containing inclusion bodies (obtained after centrifugation of Triton X-100-treated lysates), the supernatant containing solubilized proteins (prior to gel filtration FPLC), and purified VLPs—were suspended in a 2× sodium dodecyl sulfate (SDS) buffer composed of 125 mM Tris-HCl (pH 6.8), 2.0% (*w*/*v*) SDS, 0.01% bromophenol blue, 20% (*v*/*v*) glycerol, and 10% (*v*/*v*) 2-mercaptoethanol. These samples were subsequently incubated at 95 °C for 5 min. The denatured samples were then subjected to SDS-polyacrylamide gel electrophoresis (SDS-PAGE) using a 12.5% polyacrylamide gel, followed by staining with Coomassie Brilliant Blue (CBB) or subjected to Western blot analysis. CBB staining of the gel was conducted using EzStain AQua (ATTO, Tokyo, Japan).

For Western blot analysis, proteins were separated by SDS-PAGE using either a 7.5 or 12.5% acrylamide gel after heat denaturation (either at 95 °C for 5 min or at 70 °C for 10 min) in the presence of 2-mercaptoethanol or without denaturation. The separated proteins were then transferred onto a polyvinylidene difluoride (PVDF) membrane (Immobilon 0.45 μm; Merck Millipore, Tokyo, Japan). The membrane was subsequently blocked by immersion in PBS containing 0.1% (*v*/*v*) Tween-20 (PBS-T) and 5% skim milk (BD Sciences, San Jose, CA, USA), followed by thorough washing with PBS-T. The membrane was then incubated at room temperature for 1 h with an anti-HEV ORF2 mouse monoclonal antibody (MAb) (H6225 [IgG1]: 1 μg/mL) [59], which recognizes a conformational epitope (aa 551–608) of the HEV ORF2 protein and can also bind to a partially denatured epitope following heat treatment at 70 °C for 10 min [60]. After washing steps, the membrane was further incubated with horseradish peroxidase-conjugated Affinipure goat anti-mouse IgG (ProteinTech, Rosemont, IL, USA), and then visualized by chemiluminescence assay using SuperSignalTM West Atto Ultimate Sensitivity Substrate (Thermo Fisher Scientific). Protein bands were visualized using the ImageQuant LAS500 system (GE Healthcare Japan). Precision Plus Protein Dual Color Standards (Bio-Rad Laboratories, Hercules, CA, USA) served as molecular weight markers.

### 2.4. Analysis of Purified HEV VLPs with the Bioanalyzer 

The analysis of purified VLPs was conducted using the Agilent 2100 bioanalyzer coupled with the Protein 230 kit (Agilent Technologies, Palo Alto, CA, USA). Following a 1 mg/mL dilution with PBS, 4 µL of each sample was combined with 2 µL of sample buffer containing dithiothreitol. The resulting solutions were heated to 95 °C for 5 min and subsequently diluted with 84 µL of water. A total of 6 µL of each sample was then applied to the on-chip system for analysis. Molecular weight resolution across the size range of the assays was assessed utilizing protein sizing standards provided with the kits. Run control and data analysis were performed using the Agilent 2100 Expert software program (version B.02.09).

### 2.5. Transmission Electron Microscopy (TEM) 

The purified VLPs were diluted to a concentration of 0.5 mg/mL and deposited on a Formvar-coated EM grid (300 mesh) for 1 min. Subsequently, the samples underwent negative staining using a 2% uranyl acetate solution for 1 min and were analyzed utilizing a transmission electron microscope (model H-7600; Hitachi, Tokyo, Japan) operating at 80 kV.

### 2.6. Immunoelectron Microscopy (IEM)

The purified VLPs were diluted to a concentration of 0.1 mg/mL and applied onto a Formvar-coated nickel grid (300 mesh) for 20 min. Following this, PBS containing 0.2% (*w*/*v*) bovine serum albumin (BSA; Sigma-Aldrich, St. Louis, MO, USA) (PBS-0.2% BSA) was incubated on the grid for 20 min. The fixed VLPs were then exposed to an anti-HEV ORF2 MAb (H6225) (10 µg/mL in PBS-0.2% BSA) at room temperature for 1 h. After washing with PBS-0.2% BSA, the grids were treated with anti-mouse IgG conjugated with 12 nm colloidal gold (50 µg/mL in PBS-0.2% BSA; Jackson ImmunoResearch Laboratories, West Grove, PA, USA) at room temperature for 1 h. Subsequent to washing with PBS-0.2% BSA and PBS five times each, the grids were stained with a 2% uranyl acetate solution for 1 min before examination via TEM operating at 80 kV.

### 2.7. Enzyme-Linked Immunosorbent Assays (ELISAs) for the Determination of Binding with Monoclonal Antibodies (MAbs) against ORF2 Protein of Human HEV (HEV-3 and HEV-4), Rabbit HEV (HEV-3ra), or Rat HEV (HEV-C1)

Enzyme-linked immunosorbent assays (ELISAs) were conducted using 96-well microplates (Greiner Bio-One GmbH, Frickenhausen, Germany) coated with purified HEV VLPs. Briefly, 50 µL of HEV-3-VLP_112–606, HEV-3ra-VLP_112–606, and HEV-4-VLP_117–606 obtained in the present study (0.01–30 µg/mL), or rat HEV VLP (HEV-C1-VLP_357–614) (0.01–30 µg/mL) [53] diluted with PBS was added to each well. Subsequently, 100 µL of PBS containing 0.1% (*w*/*v*) BSA (PBS-0.1% BSA) was added, followed by incubation at room temperature for 1 h with shaking. After discarding the blocking buffer, the wells were washed with a washing solution containing 0.05% Tween 20 in saline. Then, 50 µL of PBS-0.1% BSA containing 0.2% Tween 20 and 1 µg/mL of anti-HEV ORF2 MAbs (H6225 or H6249 [IgG1] [59], or TA7014 [IgG2b] [61]) were added to each well, followed by another incubation at room temperature for 1 h with gentle agitation, and subsequent washing. Next, 50 µL of PBS containing 25% (*v*/*v*) fecal calf serum (FCS) (Sigma-Aldrich) and peroxidase-conjugated goat IgG fraction to mouse IgG (whole molecule) (MP Biomedicals, LLC-Cappel, Santa Ana, CA, USA) was added to each well, followed by another incubation at room temperature for 1 h with gentle agitation, and subsequent washing. Then, 50 µL of tetramethylbenzidine (TMB) soluble reagent (BioFX Laboratories, Inc., Owings Mills, MD, USA) was added to each well as a substrate. The plate was incubated at room temperature for 30 min in the dark, and then 50 µL of TMB stop buffer (BioFX Laboratories, Inc.) was added to each well. Finally, the optical density (OD) of each sample was measured at 450 nm.

### 2.8. Immunization of HEV VLPs in Mice

To evaluate the immunogenicity of HEV VLPs, three BALB/cAJcl mice each received intraperitoneal injections of 50 µg of HEV-3-VLP_112–606, HEV-3ra-VLP_112–606, or HEV-4-VLP_117–606 in complete Freund’s adjuvant (DIFCO Laboratories, Detroit, MI, USA) on day 0 and incomplete adjuvant (DIFCO Laboratories) on day 14, and serum samples were collected on day 28 at the Institute of Immunology Co., Ltd. (Tokyo, Japan). All procedures involving mice were approved by the Animal Ethics Committee of the Institute of Immunology, Co., Ltd. A pool of three non-immunized BALB/cAJcl mice served as a negative control. In addition, sera from two BALB/cAJcl mice immunized with rat HEV (HEV-C1) VLP [53] were included for comparison.

Serum samples were diluted from 3000 to 3,000,000 times with PBS-0.1% BSA and subjected to ELISA using three VLPs of HEV-3-VLP_112–606, HEV-3ra-VLP_112–606, or HEV-4-VLP_117–606 obtained in this study, rat HEV (HEV-C1) VLP_357–614 [53], or four recombinant ORF2 proteins of HEV-1, HEV-3, HEV-4 (aa 112–660) [59,62], or HEV-C1 (aa 101–644) [61] immobilized on wells of an immunoplate (Greiner Bio-One GmbH). Amino acid sequences of the immobilized proteins are depicted in Figure 2. Bound antibodies were detected using peroxidase-conjugated goat IgG fraction to mouse IgG (whole molecule) (MP Biomedicals), as described above.

## 3. Results

### 3.1. The Expression and Purification of Truncated HEV ORF2 Proteins

For HEV-3, nine types of recombinant plasmid DNAs were constructed by inserting the HEV-3 *ORF2* sequence lacking the 5′-end sequence (corresponding to aa 1–11 to 1–367 (11–367 aa)) and 3′-end sequence (corresponding to aa 607–660 (54 aa)) into the pET-3a vector (Figure 1A), and eight of nine truncated ORF2 proteins with the expected length (26–56 kDa) were found to be efficiently expressed in *E. coli* BL21 (DE3) that had been transformed with the expression plasmid DNAs and induced with IPTG; aa 12–606 proteins were expressed less efficiently (Figure 3). For HEV-3ra, two types of recombinant plasmid DNAs were constructed by inserting the HEV-3ra *ORF2* sequence lacking the 5′-end sequence (corresponding to aa 1–111 or 1–367 (111 or 367 aa)) and 3′-end sequence (corresponding to aa 607–660 (54 aa)) into the pET-3a vector (Figure 1A), and both truncated ORF2 proteins with the expected length (53 or 26 kDa) were found to be efficiently expressed in *E. coli* (Figure 3). In addition, for HEV-4, ten types of recombinant plasmid DNAs were constructed by inserting the HEV-4 *ORF2* sequence lacking the 5′-end sequence (corresponding to aa 1–111 to 1–369 (111–369 aa)) and 3′-end sequence (corresponding to aa 607–660 to 613–660 (48–54 aa)) into the pET-3a vector (Figure 1A), and all ten truncated ORF2 proteins with the expected length (26–53 kDa) were found to be efficiently expressed in *E. coli* (Figure 3).

Because the ORF2 proteins were mainly expressed in inclusion bodies in all constructs tested (Figure 3), the inclusion bodies were collected after the sonication of bacterial cells and solubilized in 4 M urea. The supernatants harvested by centrifugation of the cell lysate samples containing the target proteins were visualized by SDS-PAGE as proteins with an expected length of 26–56 kDa (Figure 4, left panels), and then subjected to gel filtration. With regard to HEV-3 ORF2 proteins of aa 12–606, 14–606, and 61–606, HEV-3ra ORF2 proteins of aa 368–606, and HEV-4 ORF2 proteins of aa 112–606, 366–606, 368–606, 370–606, 112–609, 117–609, 112–612, and 117–612, only UV peaks corresponding to hexamer and dimer fractions were discernible. In contrast, regarding HEV-3 ORF2 proteins of aa 89–606, 107–606, 112–606, 117–606, 122–606, and 368–606, HEV-3ra ORF2 proteins of aa 112–606, and HEV-4 ORF2 proteins of aa 114–606 and 117–606, a UV peak of VLP fraction was mainly found (Figure 4, right panels). These results indicated that among the expressed truncated HEV ORF2 proteins, six out of nine HEV-3 proteins, one out of two HEV-3ra proteins, and two out of ten HEV-4 proteins are capable of self-assembling into VLPs. Consequently, a total of nine HEV VLPs were successfully obtained. For subsequent analyses, solutions pooled from three to four VLP-containing fractions, as indicated by the red horizontal bar, were utilized.

### 3.2. Characterization of the Purified VLPs of HEV-3, HEV-3ra, and HEV-4

In the solutions containing purified VLPs, SDS-PAGE analysis confirmed the presence of proteins with the expected molecular weight ranges of 26 kDa and 52–56 kDa, corresponding to the monomeric forms of nine VLPs (Figure 5A). Western blot analysis of non-denatured VLPs, utilizing anti-HEV ORF2 MAb (H6225)—which specifically recognizes a conformational epitope (aa 551–608) of the HEV ORF2 protein [59,60]—revealed that three randomly selected HEV-3 VLPs (aa 89–606, 112–606, and 117–606) were detectable by MAb H6225 at molecular weights higher than those of their denatured monomeric forms. This finding confirmed that the VLPs possess the conformational epitope of the HEV ORF2 protein. Supporting this observation, the corresponding bands were undetectable by MAb H6225 when the HEV-3 VLPs (aa 89–606, 112–606, and 117–606) were heat-denatured at 95 °C for 5 min (Figure 5B). However, when the VLPs were subjected to heat treatment at 70 °C for 10 min (Figure 5C), they were detectable by MAb H6225 at molecular weights consistent with those observed in the SDS-PAGE analysis (Figure 5A), suggesting that the purified VLPs originated from the HEV ORF2 protein.

Bioanalyzer assessment revealed that the purified VLPs exhibited molecular weights ranging from 27.1 to 63.4 kDa with a purity of 74.7–95.3%. The yields of the VLPs were estimated to range from 13.9 to 89.6 mg per 100 mL of TB medium (Table 2). The reproducibility of the yields was confirmed from three independent experiments, demonstrating consistent yield and purity.

When observed with a TEM, VLPs of 20–50 (average 30) nm in diameter were visible (Figure 6, upper panels). Next, in order to confirm whether all of the observed particles of different sizes were derived from HEV ORF2, anti-HEV ORF2 MAb (TA6225), which recognizes the three-dimensional structure of HEV particles [59,60], and a mouse-IgG antibody, to which colloidal gold was conjugated, were used. IEM showed that colloidal gold was attached to VLPs of variable sizes (Figure 6, lower panels), indicating that the observed particles were derived from HEV ORF2.

### 3.3. The Antigenicity of the Purified VLPs of HEV-3, HEV-3ra, and HEV-4

The antigenicity of purified HEV VLPs was assessed using ELISAs using two anti-HEV ORF2 MAbs (H6225 and H6249) and one anti-rat HEV ORF2 MAb (TA7014). VLPs of HEV-3, HEV-3ra, and HEV-4, as well as HEV-C1 for comparison, were serially diluted from 30 μg/mL to 0.01 μg/mL, immobilized on a 96-well microplate, and detected by MAbs capable of recognizing the conformational epitope of HEV ORF2 (H6225) or rat HEV ORF2 (TA7014). These MAbs bound to HEV-3/3ra/4-VLPs or rat HEV VLPs, respectively, and detection occurred in a concentration-dependent manner (Figure 7). In contrast, VLPs of HEV-3, HEV-3ra, and HEV-4 were not detectable by an MAb against the linear epitope of HEV ORF2 protein (H6249) or an MAb against the conformational epitope of the rat HEV ORF2 protein (TA7014). These results indicate that the purified VLPs of HEV-3, HEV-3ra, and HEV-4 possess the antigenicity characteristic of HEV particles.

### 3.4. Immune Response of the Balb/cAJcl Mice to the Purified VLPs of HEV-3, HEV-3ra, and HEV-4

To evaluate the immunogenicity of the VLPs of HEV-3, HEV-3ra, and HEV-4, three Balb/cAJcl mice per group were immunized twice, on days 0 and 14, with serum samples collected on day 28. The presence of anti-HEV IgG was determined using ELISA. Four different VLPs (HEV-3, HEV-3ra, HEV-4, and HEV-C1) and four recombinant non-particulate HEV ORF2 proteins (HEV-1, HEV-3, HEV-4, and HEV-C1) were employed as the immobilized antigens. In mice immunized with the VLPs of HEV-3, HEV-3ra, or HEV-4, anti-HEV IgG was detectable using six distinct ELISAs, which utilized VLPs of HEV-3, HEV-3ra, or HEV-4, or non-particulate HEV ORF2 proteins of HEV-1, HEV-3, or HEV-4 as immobilized antigens, even after a 300,000-fold dilution (Figure 8). This suggests that HEV VLPs produced in this study exhibit high immunogenicity and can induce antibodies against both conformational and linear epitopes of the HEV ORF2 protein. In contrast, mice immunized with the VLP of HEV-C1 displayed a weak anti-HEV IgG response. However, strong anti-rat HEV IgG responses were detectable using ELISAs with HEV-C1-VLP or non-particulate HEV-C1 protein, even after a 1,000,000-fold dilution (Figure 8).

## 4. Discussion

The present study revealed that various N- and C-terminally truncated HEV ORF2 proteins (with the shortest spanning aa 370–606 and the longest spanning aa 12–606) from HEV-3 (eight out of nine proteins), HEV-3ra (both proteins), and HEV-4 (all ten proteins) were successfully expressed in *E. coli*. Among these, six HEV-3, one HEV-3ra, and two HEV-4 proteins efficiently self-assembled into VLPs. Seven of the nine purified VLPs (27.1–63.4 kDa) exhibited high purity (> 90%), with high yields of 14–90 mg per 100 mL of TB medium. The VLPs exhibited variable sizes, with diameters of 20–50 nm (average 30 nm). Western blot analysis and IEM, utilizing an anti-HEV ORF2 MAb (H6225) that recognizes the conformational epitope of the HEV ORF2 protein and neutralizes HEV infection in vitro [59,60], confirmed that the particles were derived from the HEV ORF2 protein. Their antigenicity was verified by ELISA, demonstrating that these VLPs possess characteristic HEV ORF2 antigenicity. When tested for immunogenicity in Balb/cAJcl mice, the VLPs elicited strong anti-HEV IgG responses, detectable even at high dilutions, indicating that the HEV VLPs are highly immunogenic and capable of eliciting robust immune responses against both conformational and linear epitopes of the HEV ORF2 protein. In contrast, VLPs from rat HEV (HEV-C1), obtained in our previous study [53], induced weak anti-HEV IgG but strong anti-rat HEV IgG responses (Figure 8). Overall, this study highlights the potential of these VLPs as effective immunogens for future HEV vaccine development. It also suggests the necessity of utilizing VLPs from both HEV (HEV-3, HEV-3ra, or HEV-4) and rat HEV (HEV-C1) to prevent infection from zoonotic HEV in humans.

In the initial stages of our current study, inclusion bodies in bacterial cell pellets (Section 2.2) were solubilized in 4 M urea buffer and subjected to dialysis to allow the resulting proteins to be renatured and assembled, following the method used for preparing rat HEV-VLPs in our previous study [53]. However, during dialysis, nearly all expressed proteins, including those with longer amino acid sequences, formed white turbidity, likely due to protein aggregation, making it challenging to consistently construct stable VLPs. Considering the successful development of VLPs for HEV-1_368-606 [51] and HEV-1_112-606 [52], we hypothesized that the expressed HEV ORF2 proteins of HEV-3, HEV-3ra, and HEV-4, sharing identical N- and C-terminal ends, could potentially be refolded under optimized conditions. In light of this, we explored various refolding conditions. These included stepwise dialysis of the 4 M urea-solubilized solution in buffers containing 2 M, 1 M, and 0 M urea, the addition of 0.5 M ammonium sulfate to the 4 M urea-solubilized solution prior to dialysis, and adjustments of the dialysis buffer pH from 7.5 to 8.0 or 9.0. Despite these extensive trials, stable and reproducible formation of VLPs was not achieved following dialysis and subsequent FPLC. Therefore, we omitted the dialysis step and used gel filtration to remove the urea, allowing the solubilized proteins to be renatured and assembled. This approach eliminated the need for two time-consuming operations: dialysis and ultrafiltration concentration, reducing the process time by one day. In addition, we were able to stably and reproducibly obtain at least nine VLPs of different genotypes and varying lengths (Table 2). If the refolding process occurred at a variable rate during FPLC, the resulting VLP peaks may have broadened. Although the exact mechanism responsible for the formation of the sharp peaks observed in FPLC is not yet fully understood, it is noteworthy that this phenomenon was consistently observed across all nine HEV ORF2 proteins shown in Figure 4.

The formation of VLPs is generally considered crucial for immune recognition and response [45]. Various HEV ORF2 proteins with different lengths have been expressed in *E. coli* and evaluated for their particle-forming properties. These include E2s (aa 459–606), E2 (aa 394–606), p239 (aa 368–606), and p495 (aa 112–606) of HEV-1, and p179 (aa 439–617) of HEV-4 [45,47,51,52,63,64,65]. Among them, p239 (HEV-1), p495 (HEV-1), and p179 (HEV-4) exhibited particulate characteristics. Notably, p239 (HEV-1) has been licensed as the Hecolin® recombinant VLP-based vaccine against HEV in China and Pakistan [47,51,52,54,66]. In our previous study [53], among nine rat HEV ORF2 proteins expressed in *E. coli*, five proteins of aa 357–594 (corresponding to p239 of HEV-1), 357–599, 357–604, 357–609, and 357–614 exhibited a particulate nature. Unexpectedly, the protein with the longest C-terminal extension (aa 357–614) exhibited the highest efficiency in VLP self-assembly. However, three rat HEV ORF2 proteins with longer N-termini (aa 109–614, 114–614, and 119–614) that covered all three domains (S + M + P) did not efficiently form VLPs [53], indicating the need for further investigation to determine the optimal N- and C-termini as well as specific amino acid sequences necessary for efficient VLP formation. This was particularly evident for the larger HEV VLPs corresponding to p495 (aa 112–606 of HEV-1), which were successfully expressed in *E. coli* and self-assembled [52]. In the current study, despite having identical N- and C-termini as p495 of HEV-1, both HEV-3_112–606 and HEV-3ra_112–606 formed particles, whereas HEV-4_112–606 did not. Notably, HEV-3_112–606 and HEV-4_112–606 differ by 4.2% (21/495) in the corresponding amino acid sequence. In addition, despite sharing identical N- and C-termini with p239 (Hecolin®, HEV-1), the smaller HEV VLPs showed differential particle formation; HEV-3_368–606 successfully formed particles, while both HEV-3ra_368–606 and HEV-4_368–606 did not consistently produce VLPs. Notably, HEV-3_368–606 differs from HEV-3ra_368–606 and HEV-4_368–606 by 3.3% (8/239) and 4.2% (10/239), respectively, in the amino acid sequence. These results suggest that the amino acid sequence plays a role in VLP formation in addition to the termini. For HEV-4, it was found that aa 606 was more conducive to VLP formation than aa 609 or aa 612 at the C-terminus, but it appears that the N-terminal amino acid requirements may vary by HEV strain. 

As illustrated in Figure 1A, we have constructed expression plasmids for a range of N- and C-terminally truncated ORF2 proteins, using previous studies as references and their modifications for designing the truncations. For future investigations, we plan to incorporate a theoretical prediction model to assess VLP solubility by screening for deletions. This model will be aimed at identifying key hydrophobicity-related parameters that differentiate between aggregation conducive to VLP formation and aggregation that results in insoluble viral protein clusters [67]. In the present study, our focus was not directed towards the expression of soluble proteins, as most of the proteins expressed were predominantly found in inclusion bodies (Figure 3). In future research, we plan to enhance the solubility of target proteins expressed in *E. coli* by utilizing another pET-series vector, such as pET-15b (Merck, Darmstadt, Germany), with a solubility-enhancing tag [68].

Given the highly immunogenic nature of the capsid protein, which elicits effective virus-neutralizing antibodies, and the attractive features of VLPs composed of capsid subunits—such as safety, immunogenicity, and ease of production [44,45]—HEV VLPs are promising vaccine candidates against HEV infection, as exemplified by p239 (Hecolin®) [51]. To date, HEV VLPs have been successfully produced using *E. coli* and baculovirus–insect cell systems [45]. Notably, the baculovirus–insect cell system has yielded homogeneous VLPs with diameters of ~24 nm (*T* = 1 symmetry) or ~35 nm (*T* = 3 symmetry) for HEV-1, HEV-3, HEV-4, rabbit HEV (HEV-3ra), wild boar HEV (HEV-5 and HEV-6), and camel HEV (HEV-7), as well as rat HEV and ferret HEV [50,69,70,71,72,73]. In contrast, p239-based VLPs generated using the *E. coli* system have been reported to exhibit diameters of 20–30 nm, with a degree of irregularity and heterogeneity [51]. The HEV VLPs of HEV3, HEV-3ra, and HEV-4 produced in this study using the *E. coli* expression also displayed variable sizes, ranging from 20 to 50 nm, with a mean diameter of 30 nm, resembling the rat HEV VLPs in our previous study [53], despite the differences in the lengths of the expressed proteins. The formation of uneven particle sizes after expression in *E. coli* may result from the renaturation and assembly of the ORF2 protein, which had been denatured in 4 M urea under cell-free conditions, differing from the baculovirus–insect cell system. However, despite the variability in particle size, HEV-3, HEV-3ra, and HEV-4 VLPs derived from the *E. coli* expression system in this study showed a particle profile capable of eliciting strong immunity against HEV (Figure 8). In addition, the *E. coli* expression system offers significant advantages, including rapid production within two days, compared to the baculovirus–insect cell system, which requires a minimum of two weeks [52]. Furthermore, the *E. coli* system achieves a markedly higher yield, as demonstrated in Table 2.

HEVs (HEV-1 to HEV-7) belonging to the *Paslahepevirus balayani* species have been demonstrated to represent a single serotype [72,74,75,76]. Rat HEV (HEV-C1) differs from human HEVs (HEV-1 to HEV-4) by 60–61% over the entire genome and by 55–57% in the amino acid sequence of the ORF2 protein [77]. Reflecting the differences in the amino acid sequence between rat HEV and human HEVs, rat HEV VLPs generated in the baculovirus–insect cell system showed lower cross-reactivity with sera from HEV-1-, HEV-3-, and HEV-4-infected hepatitis E patients, despite rat HEV possessing antigenic epitope(s) in common with those of HEV-1, HEV-3, and HEV-4 HEVs [69]. In addition, prior exposure to human HEV (via infection or vaccination) does not confer protection against rat HEV infection [78], suggesting the necessity of an immunogenic rat HEV vaccine to achieve effective protection against rat HEV infection in humans. The high immunogenicity of rat HEV VLPs in mice was demonstrated in our previous study [53] and confirmed in the present study (Figure 8). Further studies are warranted on the immunogenicity of a bivalent vaccine combining HEV-VLP and rat HEV-VLP in various animal species, including Mongolian gerbil, which has been reported to support infection of both human HEV and rat HEV [79,80].

In conclusion, we successfully generated self-assembled VLPs derived from HEV-3 (*n* = 6), HEV-3ra (*n* = 1), and HEV-4 (*n* = 2) strains, incorporating various N- and C-terminal modifications, through an *E. coli* expression system. These findings are expected to significantly contribute to the understanding of the antigenicity and immunogenicity of HEV, and hold promise for the development of VLP-based vaccines targeting HEV infections in humans. Future research efforts should focus on elucidating the efficient and reproducible formation of VLPs using N- and C-terminally truncated HEV ORF2 proteins of diverse genotypes, particularly those with extended amino acid sequence expressed in *E. coli*. Moreover, investigations into a bivalent vaccine strategy combining two VLPs with different immunogenic profiles are warranted.

## Figures and Tables

**Figure 1 viruses-16-01400-f001:**
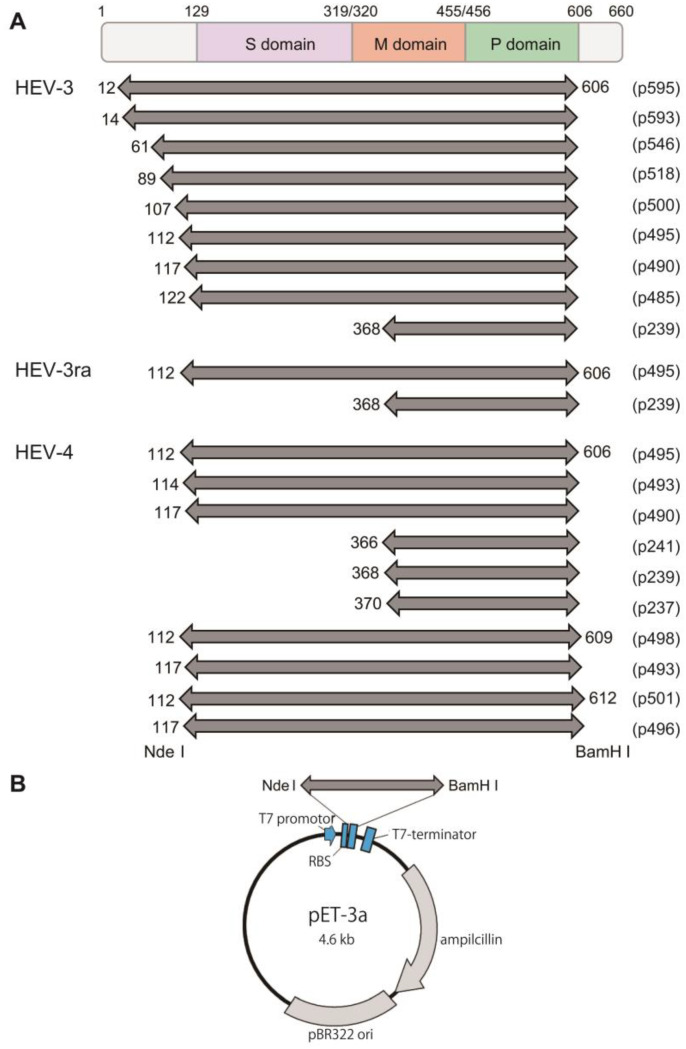
Construction of expression plasmids for truncated HEV ORF2 proteins. (**A**) A schematic diagram illustrating the three distinct domains within the ORF2 (capsid) protein: S (shell: aa 129–319), M (middle: aa 320–455), and P (protruding: aa 456–606) [57]. Nine truncated HEV-3 ORF2 proteins with varying lengths (aa 12–606, 14–606, 61–606, 89–606, 107–606, 112–606, 117–606, 122–606, and 368–606), two truncated HEV-3ra ORF2 proteins with different lengths (aa 112–606 and 368–606), and ten truncated HEV-4 ORF2 proteins with different lengths (aa 112–606, 114–606, 117–606, 366–606, 368–606, 370–606, 112–609, 117–609, 112–612, and 117–612) are depicted. Each protein’s nucleotide sequence includes Nde I and BamH I sites at the 5′- and 3′-ends, respectively. (**B**) A map of the recombinant plasmid pET-3a vector, showing the insertion of the partial HEV *ORF2* sequence at the Nde I and BamH I sites.

**Figure 2 viruses-16-01400-f002:**
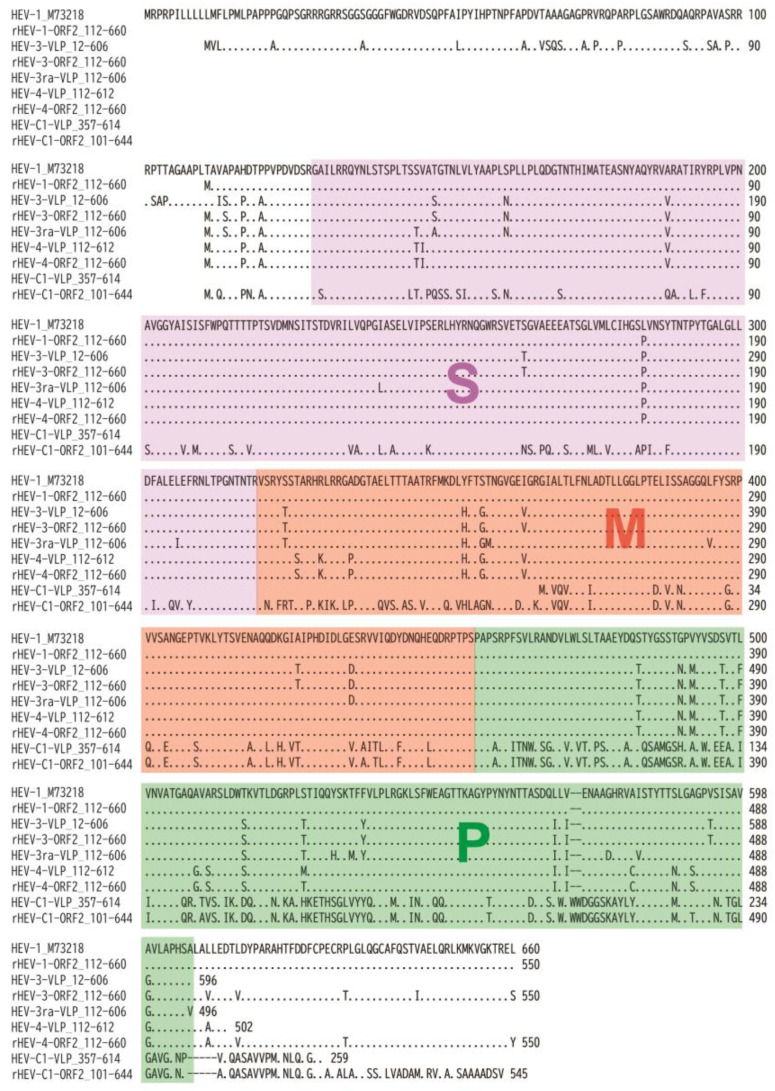
Comparison of amino acid sequences of various HEV ORF2 proteins used in this study. The 660-amino acid ORF2 sequence of the prototype HEV-1 strain (M73218) is displayed at the top. Below, the amino acid sequences of the longest HEV-3, HEV-3ra, and HEV-4 ORF2 proteins, expressed in *E. coli* in this study, are aligned. In addition, the sequences of three recombinant ORF2 proteins (aa 112–660) of HEV-1 (AB360347), HEV-3 (AB360348), and HEV-4 (AB082545) expressed in the silkworm pupae in our previous studies [58,61] are also aligned. The amino acid sequences of the rat HEV (HEV-C1) ORF2 protein, expressed in *E. coli* in our previous study [53] and in the silkworm pupae in our previous study [62], are included as well. Identical amino acids compared to the top sequence are represented by dots. The ORF2 protein’s three distinct domains—S (shell: aa 129–319), M (middle: aa 320–455), and P (protruding: aa 456–606) [57]—are highlighted in purple, red, and green, respectively.

**Figure 3 viruses-16-01400-f003:**
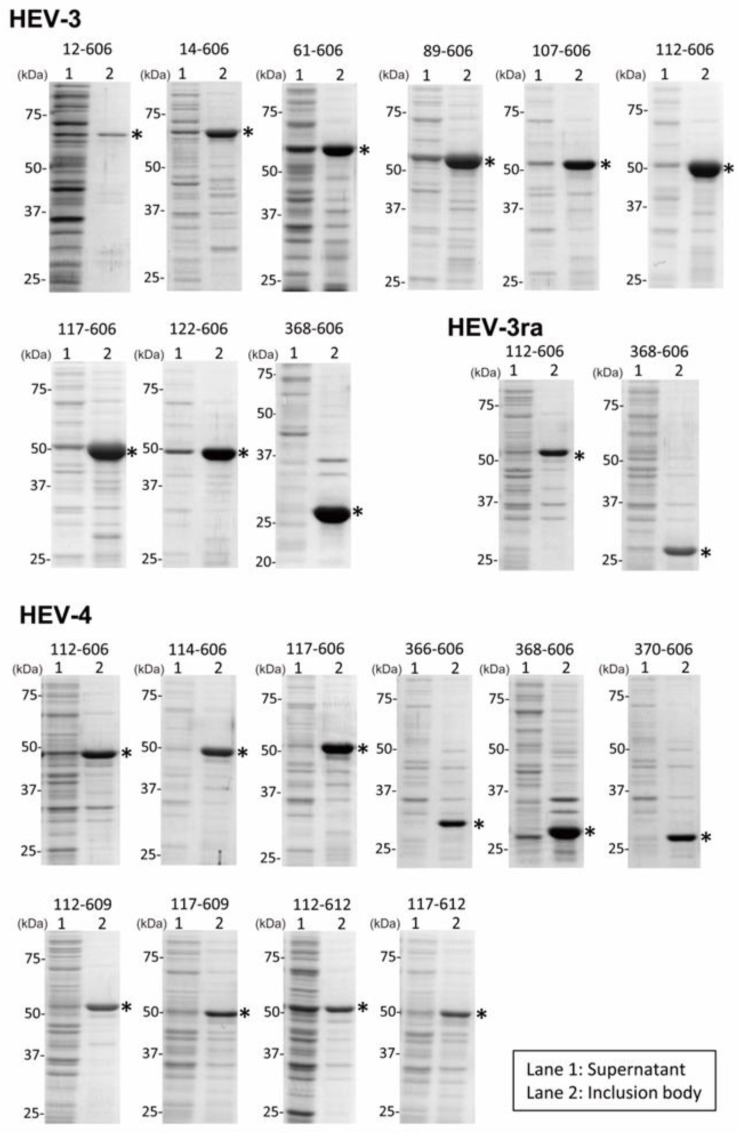
SDS-PAGE analysis of truncated HEV ORF2 proteins expressed in *E. coli*. The supernatant fraction (lane 1) and pellet fraction containing inclusion bodies (lane 2)—obtained after centrifugation of Triton X-100-treated lysates from cells expressing nine HEV-3 ORF2 proteins, two HEV-3ra ORF2 proteins, or ten HEV-4 ORF2 proteins with the indicated amino acid range—were subjected to SDS-PAGE on a 12.5% polyacrylamide gel and subsequently stained with Coomassie Brilliant Blue (CBB). One microliter of each fraction, corresponding to 5 µL of TB medium, were applied. Asterisks denote the expressed proteins of the desired length.

**Figure 4 viruses-16-01400-f004:**
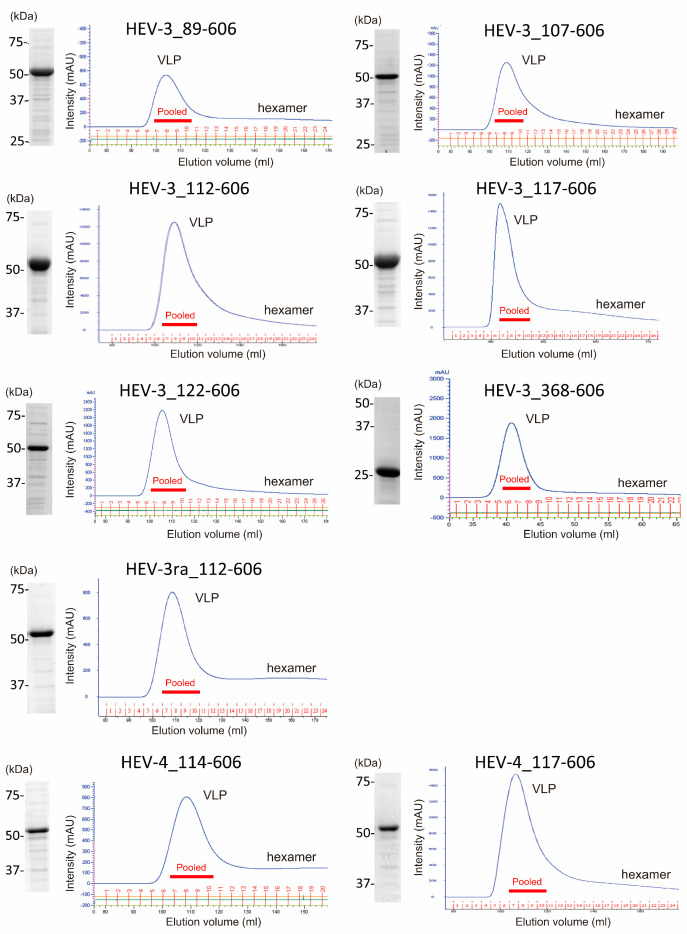
SDS-PAGE and gel filtration chromatography of HEV ORF2 proteins from sonicated and filtered solutions. The **left panel** shows the results of SDS-PAGE, while the **right panel** represents gel filtration chromatography (FPLC) data for HEV ORF2 proteins derived from sonicated and filtered solutions. These solutions were prepared in 4 M urea buffer and included six HEV-3 ORF2 proteins, one HEV-3ra ORF2 protein, and two HEV-4 ORF2 proteins. For SDS-PAGE, 1 μL of each solution was loaded on a 12.5% polyacrylamide gel and stained with Coomassie Brilliant Blue (CBB). In the gel filtration chromatography, 10 mL of solution containing the HEV ORF2 proteins, which were expressed from 33.3–100 mL of TB medium, was applied. The red horizontal bars indicate pooled fractions containing virus-like particles (VLPs) collected from three to four adjacent fractions.

**Figure 5 viruses-16-01400-f005:**
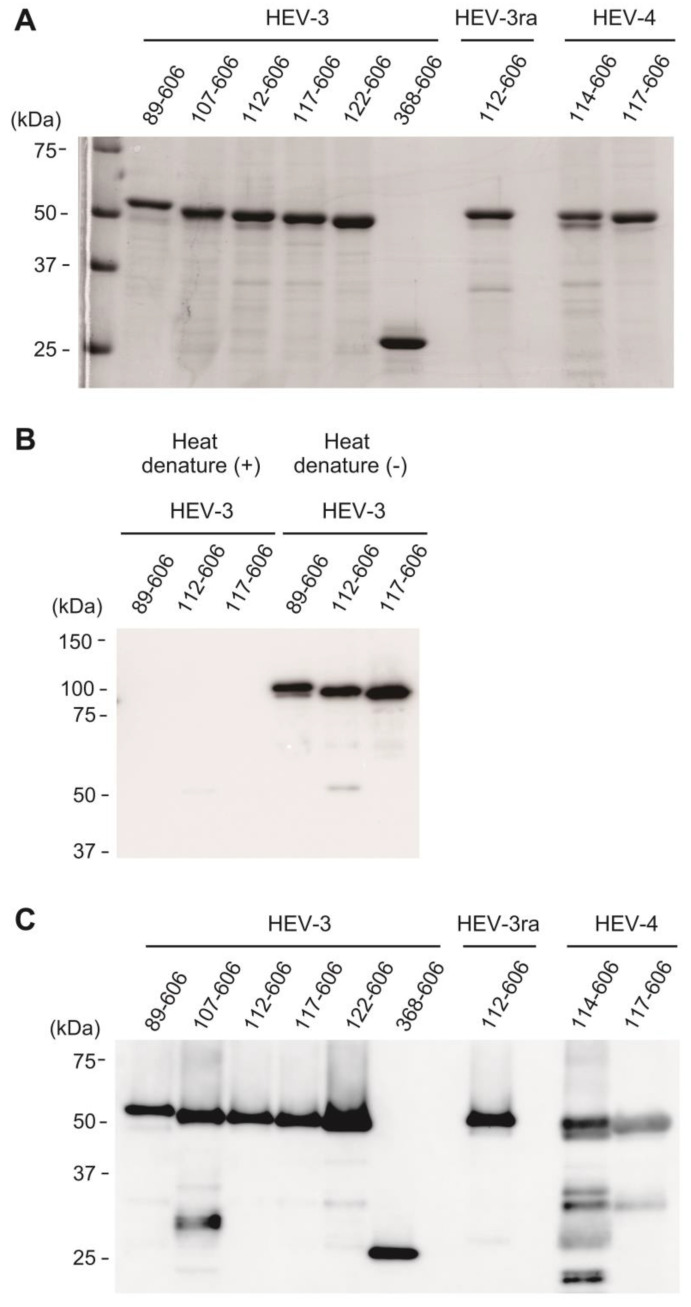
SDS-PAGE and Western blot analyses of purified virus-like particles (VLPs). (**A**) SDS-PAGE analysis of the purified virus-like particles (VLPs). A total of 4 µg of purified VLPs was heat-denatured at 95 °C for 5 min, loaded onto a 12.5% polyacrylamide gel, and stained by Coomassie Brilliant Blue (CBB). (**B**) A Western blot analysis of three representative purified VLPs using the anti-HEV ORF2 monoclonal antibody (MAb) (H6225) [59]. The samples contained 0.2 µg of purified HEV-3 VLPs (aa 89–606, 112–606, and 117–606), with or without heat denaturation at 95 °C for 5 min, and were loaded onto a 7.5% polyacrylamide gel. (**C**) A Western blot analysis of all nine purified VLPs using the anti-HEV ORF2 MAb (H6225). The samples contained 0.2 µg of purified VLPs, subjected to a partial denaturation at 70 °C for 10 min, and were loaded onto a 12.5% polyacrylamide gel. Notably, in repeated experiments, shorter additional bands, distinct from the expected band in the VLPs of HEV-3_107–606, HEV-4_114–606, and HEV-4_117–606, as observed in this figure, were likely due to protein degradation. These bands were not observed in other sample lots.

**Figure 6 viruses-16-01400-f006:**
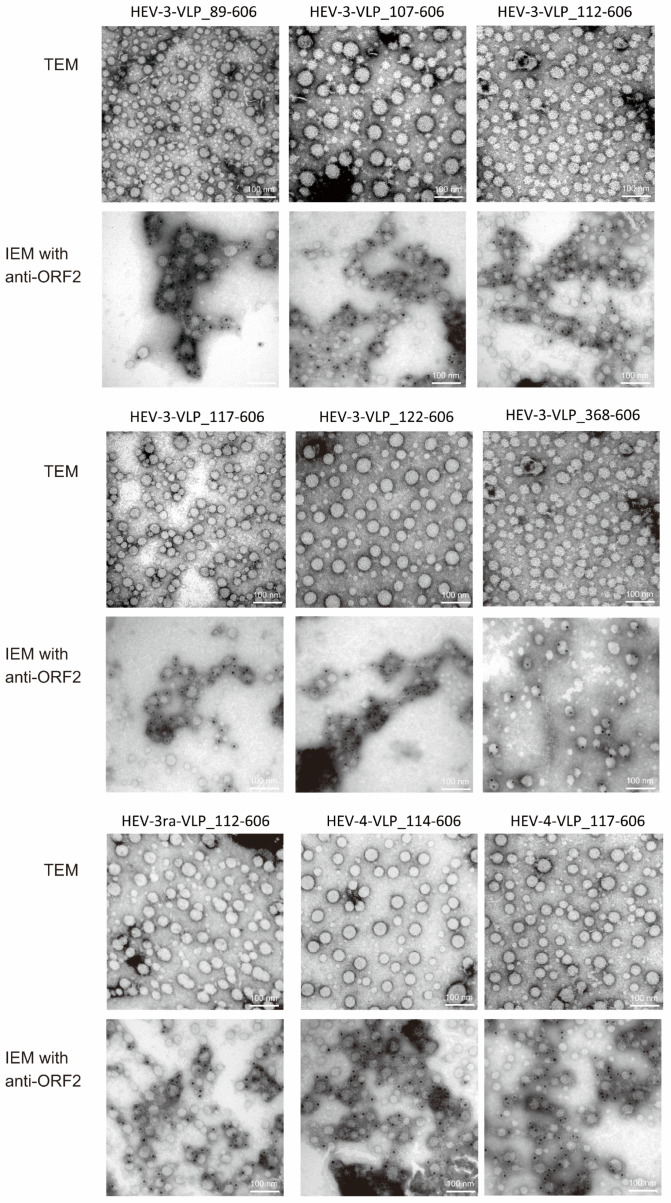
Electron micrographs of purified virus-like particles (VLPs). Electron micrographs of nine purified virus-like particles (VLPs) with the indicated genotype and amino acid range were analyzed using transmission electron microscopy (TEM, **upper panels**). The size of VLPs varied depending on the field of the electron microscope grid, with each image displaying representative VLPs averaging ~30 nm in diameter. Immunoelectron microscopy (IEM) was performed on the purified VLPs using anti-HEV ORF2 monoclonal antibody (H6225) [59] and immunogold-conjugated anti-mouse IgG (**lower panels**). The IEM images reveal heterogeneous VLPs ranging from 20 nm to 50 nm in diameter, coated with gold-labeled antibodies. Scale bar represents 100 nm.

**Figure 7 viruses-16-01400-f007:**
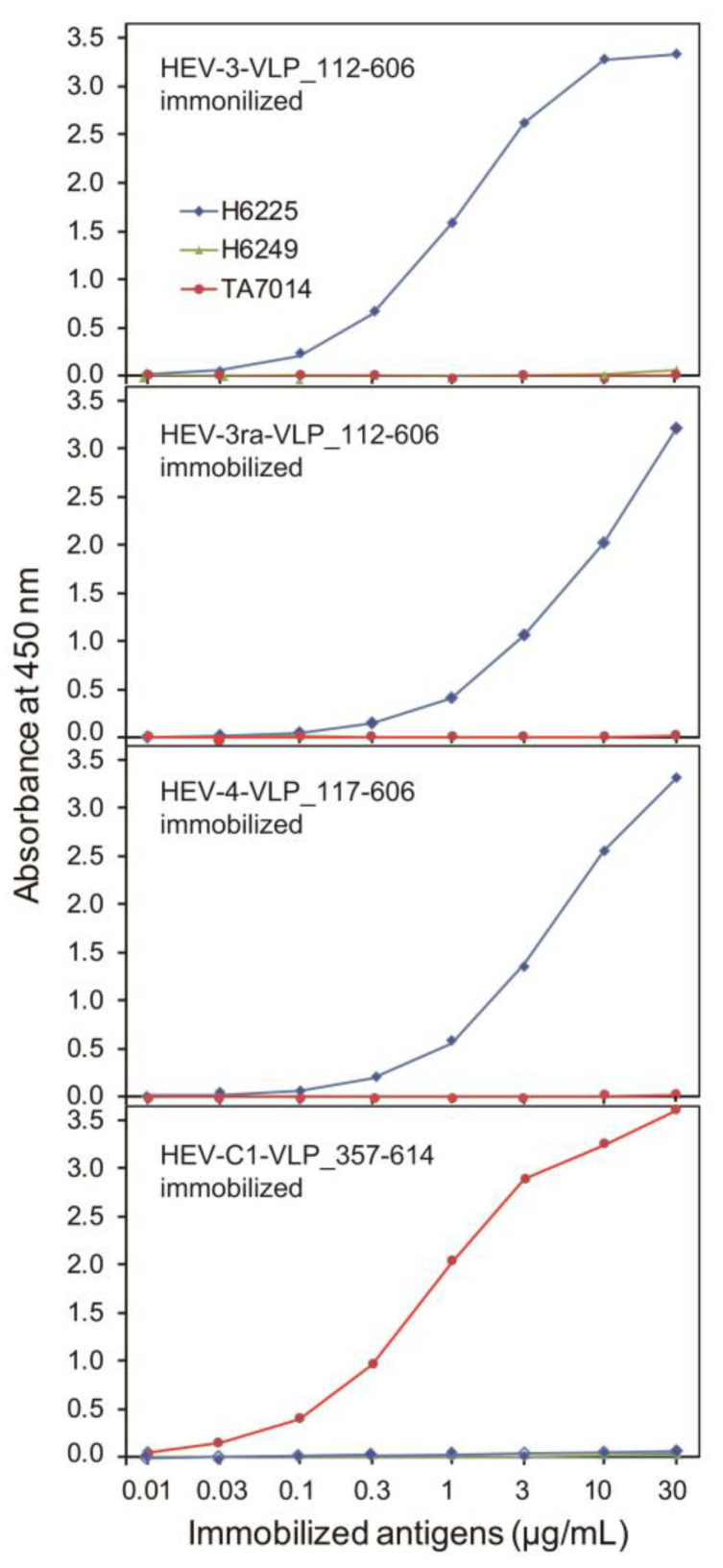
Assessment of the antigenicity of purified virus-like particles (VLPs) by enzyme-linked immunosorbent assay (ELISA). The antigenicity of HEV-3, HEV-3ra, and HEV-4 virus-like particles (VLPs) with the specified amino acid lengths was evaluated using an enzyme-linked immunosorbent assay (ELISA). Three monoclonal antibodies (MAbs) were employed in this analysis: H6225, which targets a conformational epitope of the human HEV ORF2 protein [59]; H6249, which targets a linear epitope of the human ORF2 protein [59]; and TA7014, which targets a conformational epitope of the rat HEV ORF2 protein [61]. The HEV-3, HEV-3ra, and HEV-4 VLPs were captured by MAb H6225 in a concentration-dependent manner, whereas the rat HEV (HEV-C1) VLP was captured by MAb TA7014 in a concentration-dependent manner. As a negative control, no HEV-3, HEV-3ra, or HEV4 VLPs were captured by MAb H6249.

**Figure 8 viruses-16-01400-f008:**
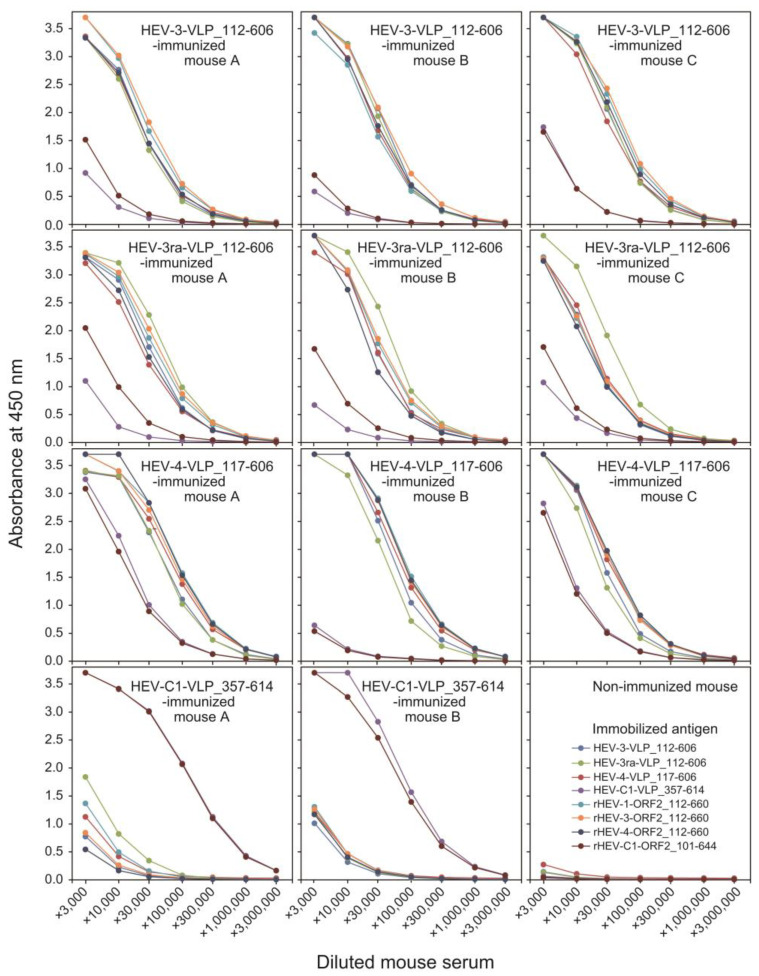
Immunogenicity assessment of antibodies against virus-like particles (VLPs) and recombinant HEV ORF2 proteins in mice immunized with VLPs. Anti-HEV IgG was detected in sera of Balb/cAJcl mice (*n* = 3 per group, designated as mouse A to C) immunized with virus-like particles (VLPs) of HEV-3, HEV-3ra, or HEV-4. Sera from mice immunized with HEV-C1 VLPs in our previous study [53] were used as a reference. Enzyme-linked immunosorbent assays (ELISAs) were conducted using VLPs (HEV-3, HEV-3ra, HEV-4, and HEV-C1) and recombinant HEV ORF2 proteins (HEV-1, HEV-3, HEV-4, and HEV-C1) as immobilized antigens. Mice immunized with HEV-3, HEV-3ra, or HEV-4 VLPs exhibited detectable anti-HEV IgG responses at a serum dilution of up to 1:300,000 across six different ELISAs. In contrast, mice immunized with HEV-C1 VLPs showed a weaker anti-HEV IgG response against the same antigen panel but exhibited a strong anti-rat HEV IgG response in ELISAs using HEV-C1 VLPs or recombinant HEV-C1 protein, detectable even at a 1,000,000-fold dilution.

**Table 1 viruses-16-01400-t001:** Primers used in this study.

Primer Name	Sequence (5′–3′)	Note
hHEV_G3 Nde-ORF2 aa12 Fw	CATATG GTGCTTCTGCCTATGCTGCC	Nde I site (underlined)
hHEV_G3 Nde-ORF2 aa14 Fw	CATATG CTGCCTATGCTGCCCGCGCC	Nde I site (underlined)
hHEV_G3 Nde-ORF2 aa61 Fw	CATATG CCCTTTGCCGCCGATGTCG	Nde I site (underlined)
hHEV_G3 Nde-ORF2 aa89 Fw	CATATG GACCAGTCCCAGCGCCCCTC	Nde I site (underlined)
hHEV_G3 Nde-ORF2 aa107 Fw	CATATG GCTGCGCCGTTGACTGCTATC	Nde I site (underlined)
hHEV_G3 Nde-ORF2 aa112 Fw	CATATG GCTATCTCACCAGCCCCTGAC	Nde I site (underlined)
hHEV_G3 Nde-ORF2 aa117 Fw	CATATG CCTGACACAGCCCCTGTACC	Nde I site (underlined)
hHEV_G3 Nde-ORF2 aa122 Fw	CATATG GTACCTGATGTTGATTCACG	Nde I site (underlined)
hHEV_G3 Nde-ORF2 aa368 Fw	CATATG ATTGCTCTGACACTGTTCAAC	Nde I site (underlined)
rbHEV_G3 Nde-ORF2 aa112 Fw	CATATG GCTGTTTCACCTGCACCTG	Nde I site (underlined)
rbHEV_G3 Nde-ORF2 aa368 Fw	CATATG ATAGCCCTGACGCTGTTTAAC	Nde I site (underlined)
hHEV_G4 Nde-ORF2 aa112 Fw	CATATG GCTGTGGCCCCGGCCCCCGA	Nde I site (underlined)
hHEV_G4 Nde-ORF2 aa114 Fw	CATATG GCCCCGGCCCCCGATACTGC	Nde I site (underlined)
hHEV_G4 Nde-ORF2 aa117 Fw	CATATG CCCGATACTGCTCCTGTTCC	Nde I site (underlined)
hHEV_G4 Nde-ORF2 aa366 Fw	CATATG GTCGGTCGTGGTATAGCGC	Nde I site (underlined)
hHEV_G4 Nde-ORF2 aa368 Fw	CATATG CGTGGTATAGCGCTAACTCTG	Nde I site (underlined)
hHEV_G4 Nde-ORF2 aa370 Fw	CATATG ATAGCGCTAACTCTGTTCAATC	Nde I site (underlined)
hHEV_G3 ORF2-Bam aa606 Rev	GGATCC CTA AGCCGAGTGCGGGGCTAGTAC	BamH I site (underlined)
rbHEV_G3 ORF2-Bam aa606 Rev	GGATCC CTA AACTGAGTGCGGAGCAAGC	BamH I site (underlined)
hHEV_G4 ORF2-Bam aa606 Rev	GGATCC CTA CGCAGAGTGAGGTGCGAGGAC	BamH I site (underlined)
hHEV_G4 ORF2-Bam aa609 Rev	GGATCC CTA AGCGGCCAGCGCAGAGTGAG	BamH I site (underlined)
hHEV_G4 ORF2-Bam aa612 Rev	GGATCC CTA GTCCTCTAAAGCGGCCAGCG	BamH I site (underlined)
rbHEV_G3 T6686C Fw	TACGGCTCGTCAACTAACCC	nt 6684-6703 ^a^ (mutated nt underlined)
rbHEV_G3 T6686C Rev	TGTAGTTTGATCATACTCTG	nt 6664-6683 ^a^

^a^ Nucleotide positions are numbered in accordance with the rbIM223L strain (LC775585).

**Table 2 viruses-16-01400-t002:** The yields of purified virus-like particles (VLPs) expressed in 100 mL of TB medium (*E. coli*).

Purified VLPs	Amount (mL) Recovered after Gel Filtration	Concentration (Mean ± SD, mg/mL) ^a^	Molecular Weight (Mean ± SD, kDa) ^b^	Yield (Mean ± SD, mg)	Purity (Mean ± SD, %) ^b^
HEV-3-VLP_89-606 (p518)	48	0.5 ± 0.1	63.4 ± 0.7	24.0 ± 6.8	89.5 ± 2.8
HEV-3-VLP_107-606 (p500)	48	1.4 ± 0.1	61.1 ± 0.4	66.0 ± 6.0	90.7 ± 4.1
HEV-3-VLP_112-606 (p495)	48	1.9 ± 0.3	61.0 ± 0.5	89.6 ± 15.4	95.3 ± 1.4
HEV-3-VLP_117-606 (p490)	48	1.3 ± 0.1	60.3 ± 0.6	64.0 ± 5.5	91.9 ± 2.3
HEV-3-VLP_122-606 (p485)	48	1.6 ± 0.4	58.4 ± 1.1	76.8 ± 17.3	95.0 ± 1.6
HEV-3-VLP_368-606 (p239)	12	1.4 ± 0.6	27.1 ± 0.5	16.2 ± 7.6	93.1 ± 0.1
HEV-3ra-VLP_112-606 (p495)	16	0.9 ± 0.2	62.7 ± 0.6	13.9 ± 3.7	94.9 ± 1.9
HEV-4-VLP_114-606 (p493)	48	0.8 ± 0.1	61.9 ± 0.7	39.2 ± 5.0	74.7 ± 5.8
HEV-4-VLP_117-606 (p490)	48	1.3 ± 0.4	60.7 ± 0.3	62.4 ± 20.4	90.1 ± 8.5

^a^ Mean ± standard deviation (SD) protein concentration values, quantified using the NanoDrop 2000c spectrophotometer (Thermo Fisher Scientific, Waltham, MA, USA) from three independent experiments, are indicated. ^b^ Mean ± SD molecular weight or purity values, determined using the Agilent 2100 bioanalyzer (Agilent Technologies, Palo Alto, CA, USA) from three independent experiments, are indicated.

## Data Availability

All data generated or analyzed during this study are either incorporated within this published article or accessible upon reasonable request to the corresponding author. The nucleotide sequence data reported in this study have been assigned the DDBJ/EMBL/GenBank accession numbers LC775584 and LC775585.

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
