# Peer review of "Production and Characterization of Self-Assembled Virus-like Particles Comprising Capsid Proteins from Genotypes 3 and 4 Hepatitis E Virus (HEV) and Rabbit HEV Expressed in Escherichia coli"

_viruses, 2024, doi:10.3390/v16091400_

Round 1

Reviewer 1 Report

Comments and Suggestions for Authors

This manuscript reports success in a challenging area, i.e. making VLP (virus-like particles) using recombinant animal virus (HEV) antigenic proteins produced in E. coli. The authors took the laborious ask of expressing a large body of nested deletions of the proteins, and making VLPs out of proteins that were expressed in inclusion bodies and needed to be dissolved by urea. Several of the VLPs were found to be antigenic, and therefore, useful for vaccination. The ultimate success came from hard work, based on experience and large amounts of screening by the multimember team. The results are very clean and beautiful.

My main criticism is it lacks some rational reasoning for the empirical experiments, and I provide the specific suggestions below:

1) State the criterion of a calling a protein complex VLP. How does the round structures seen under immnoelectron microscopy meets this definition. Do these particles also mimic the icosahedral structure of the virion?

2) Please explain the statement “The formation of VLPs is generally considered crucial for immune recognition and response” (Line 501). Any citation, or explanation as to why?

3) Did the authors screen the deletions for theoretical prediction of solubility before the deletions were actually done, for example by hydropathicity plot using available algorithms (Kyle-Doolittle, Hop-Woods etc)

4) It appears that the authors did not try much to express a soluble protein. For example, did they try using a pET-series vector with a tag such as pET-15b, some of which offer cleavable tags)?  Novagen made many kinds of pET vectors (now marketed and further improved by EMD-Millipore).  The ligation-independent vectors do not require the use of a restriction site.

Comments on the Quality of English Language

Minor improvements needed.

Author Response

Responses to the comments from Reviewer #1:

“This manuscript reports success in a challenging area, i.e. making VLP (virus-like particles) using recombinant animal virus (HEV) antigenic proteins produced in E. coli. The authors took the laborious ask of expressing a large body of nested deletions of the proteins, and making VLPs out of proteins that were expressed in inclusion bodies and needed to be dissolved by urea. Several of the VLPs were found to be antigenic, and therefore, useful for vaccination. The ultimate success came from hard work, based on experience and large amounts of screening by the multimember team. The results are very clean and beautiful.”

  RESPONSE: Thank you for your favorable comments and kind suggestions to improve our manuscript.

“My main criticism is it lacks some rational reasoning for the empirical experiments, and I provide the specific suggestions below:

  • State the criterion of a calling a protein complex VLP. How does the round structures seen under immnoelectron microscopy meets this definition. Do these particles also mimic the icosahedral structure of the virion?”

RESPONSE: The VLPs generated in the present study demonstrated strong antigenic reactivity with an MAb H6225, which specifically recognizes the conformational epitope of the HEV ORF2 protein. This antibody is known for its ability to capture fecal HEV particles and possesses neutralizing activity. This reactivity was confirmed through enzyme-linked immunosorbent assay (ELISA) (Fig. 7) and Western blotting (Figs. 5B and 5C). Additionally, these VLPs elicited a robust anti-HEV IgG response in a mouse immunization assay, detectable at serum dilutions as high as 1:300,000 (Fig. 8). Transmission electron microscopy (TEM) and immunoelectron microscopy (IEM) using the anti-ORF2 MAb H6225, along with colloidal gold labeling, revealed the presence of round structures associated with the VLPs (Fig. 6). These findings suggest that the observed particles with round structures satisfy the criteria for classification as VLPs. However, due to the size variability observed (20–50 nm, mean 30 nm), we were unable to conclusively determine whether these particles mimic the icosahedral structure of the virion.

  • “Please explain the statement “The formation of VLPs is generally considered crucial for immune recognition and response” (Line 501). Any citation, or explanation as to why?”

 RESPONSE: Thank you for your comment. The relevant reference (Ref. 45) was cited in the sentence (Line 515).

  • “Did the authors screen the deletions for theoretical prediction of solubility before the deletions were actually done, for example by hydropathicity plot using available algorithms (Kyle-Doolittle, Hop-Woods etc).”

 RESPONSE: As illustrated in Fig. 1A, we have constructed expression plasmids for a range of N- and C-terminally truncated ORF2 proteins, using previous studies as references and their modifications for designing the truncations. For future investigations, we plan to incorporate a theoretical prediction model to assess VLP solubility by screening for deletions. This model will be aimed at identifying key hydrophobicity-related parameters that differentiate between aggregation conducive to VLP formation and aggregation that results in insoluble viral protein clusters (Lines 543-549).

  • “It appears that the authors did not try much to express a soluble protein. For example, did they try using a pET-series vector with a tag such as pET-15b, some of which offer cleavable tags)?  Novagen made many kinds of pET vectors (now marketed and further improved by EMD-Millipore).  The ligation-independent vectors do not require the use of a restriction site.”

 RESPONSE: We appreciate your insightful observation regarding our limited exploration of soluble protein expression in E. coli within our current study. In future research, we intend to pursue the expression of soluble proteins in E. coli using the recommended expression vectors, including another pET-series vector with a tag such as pET-15b (Merck, Darmstadt, Germany), to enhance the solubility of our target proteins. The following sentences were added to the Discussion section: In the present study, our focus was not directed towards the expression of soluble proteins, as most of the proteins expressed were predominantly found in inclusion bodies. In future research, we plan to enhance the solubility of target proteins expressed in E. coli by utilizing another pET-series vector, such as pET-15b (Merck, Darmstadt, Germany), with a solubility-enhancing tag [76] (Lines 549-553).

Reviewer 2 Report

Comments and Suggestions for Authors

A comprehensive investigation of the recombinant expression of HEV-3, HEV-3a, and HEV-4 ORF2 in E. coli is presented by Kobayashi et al. Their research is important for the development of a hepatitis E vaccine that covers a wider range of HEV genotypes. Furthermore, the authors proposed an effective method for ORF2 capsid protein purification and VLPs preparation. I congratulate the authors for their excellent scientific development.

There are only some technical notes about the introduction and the cited literature.

L27, 29, 31, 99, 100, 109, 169, 502,517, 534, 539,541, 548, 544,549, 552, 571  E.coli, and all Latin names must be in Italic.

L28 However, research on the production of ORF2 proteins from these HEV genotypes in E. coli to form VLPs is limited. Please, rewrite the sentence, there are examples of HEV-3 and HEV-4 ORF2 expression in E. coli. These studies are not as complete as your research on the structure of the recombinant ORF2, but the examples are not limited.  Your statement is only valid for HEV-3ra.

The introduction is written quite nicely, however a few of the citations are incorrectly used.

L 100
Please, do not cite the review papers [45, 46], but the research articles themselves.

Robinson, R.A.; Burgess, W.H.; Emerson, S.U.; Leibowitz, R.S.; Sosnovtseva, S.A.; Tsarev, S.; Purcell, R.H. Structural Characterization of Recombinant Hepatitis E Virus ORF2 Proteins in Baculovirus-Infected Insect Cells. Protein Expr. Purif. 199812, 75–84

Li, S.W.; Zhang, J.; Li, Y.M.; Ou, S.H.; Huang, G.Y.; He, Z.Q.; Ge, S.X.; Xian, Y.L.; Pang, S.Q.; Ng, M.H.; et al. A Bacterially Expressed Particulate Hepatitis E Vaccine: Antigenicity, Immunogenicity and Protectivity on Primates. Vaccine 200523, 2893–2901. 

Dong, C.; Meng, J. Expression, purification and immunogenicity of a novel hepatitis E virus-like particle. Xi Bao Yu Fen Zi Mian Yi Xue Za Zhi Chin. J. Cell. Mol. Immunol. 200622, 339–342.

Zahmanova, G.G.; Mazalovska, M.; Takova, K.H.; Toneva, V.T.; Minkov, I.N.; Mardanova, E.S.; Ravin, N.V.; Lomonossoff, G.P. Rapid High-Yield Transient Expression of Swine Hepatitis E ORF2 Capsid Proteins in Nicotiana Benthamiana Plants and Production of Chimeric Hepatitis E Virus-Like Particles Bearing the M2e Influenza Epitope. Plants 20209, 29

The HEV-4 ORF2 p179 (aa 439–617) protein product, expressed in E. coli, was developed by Changchun Institute of Biological Products Co., Ltd. (Changchun, China) and passed the phase I clinical trial (Clinical trial NO. CXSL1000041) Cao et al. Vaccine 201735, 5073–508

L107 - Notably, the construction of VLPs from HEV-3 including HEV-3ra (rabbit HEV), has not been reported. This data is not fully correct (Simanavicius et al.,  Appl. Microbiol. Biotechnol. 2018102, 185–198; Zahmanova et al., Plants 2020 reported VLPs formation from HEV-3 ORF2). Please, rewrite the sentence and include this information

L 71, 72, 73 Hepeviridae, Orthohepevirinae and Parahepevirinae, Paslahepevirus, Rocahepevirus, Chirohepevirus, and Avihepevirus must be in Italic.

L91 - Rocahepevirus ratti – Italic

Materials and methods are described in detail.

L191 - 0.45 m pore-size please, check

I congratulate you on your good results and neat logical presentation.

Fig. 5 (B) A Western blot analysis of three representative purified VLPs using the 380

anti-HEV ORF2 monoclonal antibody (MAb) (H6225) [58]. Please, explain the type of MAb) (H6225), and why we do not observe signals after denaturation of proteins.

L520 –  I recommend  the abbreviation aa be replaced with the full name

L530, The reference articles used are review articles, making it very general. Use specific researchers' articles for references.  Please change references 44, 45 with more reliable.

L534 This information is repeated

L540 HEV3 check it

L530-552 – It would be interesting to comment on other expression systems for HEV VLPs production, the VLPs heterogenesity, and the role of posttranslational modification for VLPs formation 

L553 Paslahepevirus balayani Italic

L569 Conclusion, make it separate chapter 5

L572 These findings are poised to advance  studies on HEV antigenicity, epidemiology and pathogenicity, and hold promise for the development of VLP-based vaccines targeting HEV infections in humans. Please, rewrite the sentence, your main discoveries are not aimed at the epidemiology and pathogenicity of HEV. 

Author Response

Responses to the comments from Reviewer #2:

“A comprehensive investigation of the recombinant expression of HEV-3, HEV-3a, and HEV-4 ORF2 in E. coli is presented by Kobayashi et al. Their research is important for the development of a hepatitis E vaccine that covers a wider range of HEV genotypes. Furthermore, the authors proposed an effective method for ORF2 capsid protein purification and VLPs preparation. I congratulate the authors for their excellent scientific development.

There are only some technical notes about the introduction and the cited literature.”

RESPONSE: Thank you for your positive comments and thorough review for improving our manuscript.

“L27, 29, 31, 99, 100, 109, 169, 502,517, 534, 539,541, 548, 544,549, 552, 571  E.coli, and all Latin names must be in Italic.”

  RESPONSE: In accordance with your kind suggestions, all words that must be in italic in the manuscript were italicized.

“L28 However, research on the production of ORF2 proteins from these HEV genotypes in E. coli to form VLPs is limited. Please, rewrite the sentence, there are examples of HEV-3 and HEV-4 ORF2 expression in E. coli. These studies are not as complete as your research on the structure of the recombinant ORF2, but the examples are not limited.  Your statement is only valid for HEV-3ra.”

RESPONSE: Thank you for your thoughtful comment. The statement was rewritten to “However, research on the production of ORF2 proteins from these HEV genotypes in E. coli to form VLPs has been modest” (Lines 28-29).

“The introduction is written quite nicely, however a few of the citations are incorrectly used.
L 100 Please, do not cite the review papers [45, 46], but the research articles themselves.

Robinson, R.A.; Burgess, W.H.; Emerson, S.U.; Leibowitz, R.S.; Sosnovtseva, S.A.; Tsarev, S.; Purcell, R.H. Structural Characterization of Recombinant Hepatitis E Virus ORF2 Proteins in Baculovirus-Infected Insect Cells. Protein Expr. Purif. 1998, 12, 75–84

Li, S.W.; Zhang, J.; Li, Y.M.; Ou, S.H.; Huang, G.Y.; He, Z.Q.; Ge, S.X.; Xian, Y.L.; Pang, S.Q.; Ng, M.H.; et al. A Bacterially Expressed Particulate Hepatitis E Vaccine: Antigenicity, Immunogenicity and Protectivity on Primates. Vaccine 2005, 23, 2893–2901. 

Dong, C.; Meng, J. Expression, purification and immunogenicity of a novel hepatitis E virus-like particle. Xi Bao Yu Fen Zi Mian Yi Xue Za Zhi Chin. J. Cell. Mol. Immunol. 2006, 22, 339–342.

Zahmanova, G.G.; Mazalovska, M.; Takova, K.H.; Toneva, V.T.; Minkov, I.N.; Mardanova, E.S.; Ravin, N.V.; Lomonossoff, G.P. Rapid High-Yield Transient Expression of Swine Hepatitis E ORF2 Capsid Proteins in Nicotiana Benthamiana Plants and Production of Chimeric Hepatitis E Virus-Like Particles Bearing the M2e Influenza Epitope. Plants 2020, 9, 29

The HEV-4 ORF2 p179 (aa 439–617) protein product, expressed in E. coli, was developed by Changchun Institute of Biological Products Co., Ltd. (Changchun, China) and passed the phase I clinical trial (Clinical trial NO. CXSL1000041) Cao et al. Vaccine 2017, 35, 5073–508”

  RESPONSE: We sincerely appreciate your valuable suggestion. However, we would like to note that the article by Dong et al. (Xi Bao Yu Fen Zi Mian Yi Xue Za Zhi, Chin. J. Cell. Mol. Immunol. 2006) is published in Chinese, and the remaining four articles are appropriately cited in the review article referenced as Ref. 44. Therefore, we kindly request to retain the citation of the two most relevant review articles (Refs. 44 and 45) in the Introduction section.

“L107 - Notably, the construction of VLPs from HEV-3 including HEV-3ra (rabbit HEV), has not been reported. This data is not fully correct (Simanavicius et al.,  Appl. Microbiol. Biotechnol. 2018102, 185–198; Zahmanova et al., Plants 2020 reported VLPs formation from HEV-3 ORF2). Please, rewrite the sentence and include this information”

RESPONSE: Thank you for your valuable input. Our original statement was specifically intended to refer to the construction of VLPs from HEV-3, including HEV-3ra (rabbit HEV), in E. coli. To clarify this point, we have revised the sentence to read: "Notably, the construction of VLPs from HEV-3, including HEV-3ra (rabbit HEV), in E. coli has not been reported." (Lines 107-108).

“L 71, 72, 73 Hepeviridae, Orthohepevirinae and Parahepevirinae, Paslahepevirus, Rocahepevirus, Chirohepevirus, and Avihepevirus must be in Italic.

L91 - Rocahepevirus ratti – Italic”

RESPONSE: As suggested, all these words were italicized.

“Materials and methods are described in detail.”

  RESPONSE: Thank you for your positive comment.

“L191 - 0.45 m pore-size please, check”

RESPONSE: “0.45 m” was revised to “0.45 mm” (Line 190).

“I congratulate you on your good results and neat logical presentation.”

  RESPONSE: Thank you for your favorable comment.

“Fig. 5 (B) A Western blot analysis of three representative purified VLPs using the 380

anti-HEV ORF2 monoclonal antibody (MAb) (H6225) [58]. Please, explain the type of MAb) (H6225), and why we do not observe signals after denaturation of proteins.”

  RESPONSE: As described in the revised manuscript (Line xxx), MAb H6225 is of the IgG1 subclass. Please note that since MAb H6225 recognizes the conformation epitope of the HEV ORF2 protein, it cannot bind to the linear epitope of the heat-denatured ORF2 protein.

“L520 –  I recommend  the abbreviation aa be replaced with the full name”

  RESPONSE: “aa” was replaced with the full name “amino acid”.

“L530, The reference articles used are review articles, making it very general. Use specific researchers' articles for references.  Please change references 44, 45 with more reliable.”

RESPONSE: We sincerely appreciate your insightful suggestion. However, we kindly request to retain the citations of the two most relevant and up-to-date review articles (Refs. 44 and 45). These reviews provide comprehensive coverage of the topic, with one of them being published in the highly regarded journal Viruses.

“L534 This information is repeated”

  RESPONSE: Please note that this statement is not repeated in the Discussion section.

“L540 HEV3 check it”

RESPONSE: As suggested, HEV3 was revised to HEV-3 (Line 564).

“L530-552 – It would be interesting to comment on other expression systems for HEV VLPs production, the VLPs heterogenesity, and the role of posttranslational modification for VLPs formation” 

RESPONSE: We appreciate your insightful suggestion to explore other expression systems for HEV VLP production, as well as the VLP heterogeneity and the role of post-translational modifications in VLP formation. While these aspects are indeed intriguing, they fall outside the scope of the current study.

“L553 Paslahepevirus balayani Italic”

RESPONSE: As suggested, the word was italicized.

“L569 Conclusion, make it separate chapter 5”

RESPONSE: Since it is not required by the journal, we would like to retain “the sentences in conclusion” as it stands.

“L572 These findings are poised to advance  studies on HEV antigenicity, epidemiology and pathogenicity, and hold promise for the development of VLP-based vaccines targeting HEV infections in humans. Please, rewrite the sentence, your main discoveries are not aimed at the epidemiology and pathogenicity of HEV.” 

  RESPONSE: In accordance with your valuable suggestion, the sentence was rewritten to “These findings are expected to significantly contribute to the understanding of the antigenicity and immunogenicity of HEV, and hold promise for the development of VLP-based vaccines targeting HEV infections in humans.” (Lines 595-598).

Reviewer 3 Report

Comments and Suggestions for Authors

HEV is a major cause of acute liver disease and can cause huge outbreaks in developing countries.  A vaccine is available but is marketed in only 2 countries.  Zoonotic transmission to humans is becoming more recognized, raising the need for greater vaccine preparedness.  Here, Kobayasi and colleagues adapt and extend HEV VLP production technologies and demonstrate 3 new VLP preparations from human genotypes not in the existing vaccines.

Many strengths are evident in the manuscript.  It is very straight-forward experimentally, the data support the authors’ conclusions, it is well written, it reports a new way to re-fold bacterially produced HEV E2 proteins into VLPs, and it demonstrates that the new VLPs have the appropriate immunoreactivity and immunogenicity.  These VLPs will be useful for characterizing genotype-specificity of vaccines (if it exists), and could form the starting point for novel vaccines should they become necessary.

Weaknesses are small.

1)  The refolding procedure is inadequately described.  It was apparently done during SEC, which raises the question as to why the peaks are so sharp in Fig. 4 if folding proceeded at a heterogeneous pace during chromatography.  Specifically, how much was loaded and what was the temperature and flow rate? 

2)  Were only 2 refolding conditions tried?  Do the authors anticipate that some of the constructs that did not form VLPs could be induced to do so if conditions were optimized? 

Small item:

Sometimes the units were unclear in my PDF—perhaps the µ symbol was not converted properly during PDF generation?

Author Response

Responses to the comments from Reviewer #3:

“HEV is a major cause of acute liver disease and can cause huge outbreaks in developing countries.  A vaccine is available but is marketed in only 2 countries.  Zoonotic transmission to humans is becoming more recognized, raising the need for greater vaccine preparedness.  Here, Kobayasi and colleagues adapt and extend HEV VLP production technologies and demonstrate 3 new VLP preparations from human genotypes not in the existing vaccines.

Many strengths are evident in the manuscript.  It is very straight-forward experimentally, the data support the authors’ conclusions, it is well written, it reports a new way to re-fold bacterially produced HEV E2 proteins into VLPs, and it demonstrates that the new VLPs have the appropriate immunoreactivity and immunogenicity.  These VLPs will be useful for characterizing genotype-specificity of vaccines (if it exists), and could form the starting point for novel vaccines should they become necessary.

Weaknesses are small.”

  RESPONSE: Thank you for your favorable evaluation and thorough review for improving our manuscript.

  • “The refolding procedure is inadequately described.  It was apparently done during SEC, which raises the question as to why the peaks are so sharp in Fig. 4 if folding proceeded at a heterogeneous pace during chromatography.  Specifically, how much was loaded and what was the temperature and flow rate?” 

RESPONSE: We greatly appreciate your insightful comments. In response, we would like to clarify that 11 milliliters of filtered supernatant were loaded onto the FPLC column at 4°C, with a flow rate ranging from 1.0 to 1.4 ml/min. As a result, three to four fractions (12 to 16 ml) containing VLPs were collected within 30 to 40 minutes post-loading (Lines 192-198). While the precise mechanism underlying the formation of the sharp peaks in FPLC remains unclear, it is important to note that this phenomenon was consistently observed across all nine HEV ORF2 proteins presented in Fig. 4 (Lines 509-513).

2)  “Were only 2 refolding conditions tried?  Do the authors anticipate that some of the constructs that did not form VLPs could be induced to do so if conditions were optimized? “

  RESPONSE: Considering the successful development of VLPs for HEV-1_368-606 [51] and HEV-1_112-606 [52], we hypothesized that the expressed HEV ORF2 proteins of HEV-3, HEV-3ra, and HEV-4, sharing identical N- and C-terminal ends, could potentially be refolded under optimized conditions. In light of this, we explored various refolding conditions. These included stepwise dialysis of the 4 M urea-solubilized solution in buffers containing 2 M, 1 M, and 0 M urea, the addition of 0.5 M ammonium sulfate to the 4 M urea-solubilized solution prior to dialysis, and adjustments of the dialysis buffer pH from 7.5 to 8.0 or 9.0. Despite these extensive trials, stable and reproducible formation of VLPs was not achieved following dialysis and subsequent FPLC (Lines 496-504).

“Small item:

Sometimes the units were unclear in my PDF—perhaps the µ symbol was not converted properly during PDF generation?”

  RESPONSE: Thank you for your kind suggestions. Various symbols that were not converted properly during PDF generation were carefully added or revised throughout the manuscript.
